# The Effectiveness of Nutrition Education for Overweight/Obese Mother with Stunted Children (NEO-MOM) in Reducing the Double Burden of Malnutrition

**DOI:** 10.3390/nu10121910

**Published:** 2018-12-04

**Authors:** Trias Mahmudiono, Abdullah Al Mamun, Triska Susila Nindya, Dini Ririn Andrias, Hario Megatsari, Richard R. Rosenkranz

**Affiliations:** 1Department of Nutrition, Faculty of Public Health, Universitas Airlangga, Surabaya 60115, Indonesia; triskasnindya@yahoo.com (T.S.N.); dien_ra@yahoo.com (D.R.A.); 2Southeast Asian Ministers of Education Organization Regional Center for Food and Nutrition (SEAMEO RECFON), Pusat Kajian Gizi Regional (PKGR), Universitas Indonesia, Jakarta 10430, Indonesia; 3Institute for Social Science Research, The University of Queensland, Indooroopilly, Queensland 4068, Australia; mamun@sph.uq.edu.au; 4Department of Health Promotion and Behavior Sciences, Faculty of Public Health, Universitas Airlangga, Surabaya 60115, Indonesia; hario.megatsari@gmail.com; 5Department of Food, Nutrition, Dietetics and Health, Kansas State University, Manhattan, KS 66506, USA; ricardo@ksu.edu

**Keywords:** nutrition education, health promotion, behavioral intervention, self-efficacy, stunting, overweight, obesity, physical activity, dual burden of malnutrition

## Abstract

(1) Background: In households experiencing the double burden of malnutrition, stunted children are in a better position for growth improvement when parents are able to direct their resources to support nutrition requirements. This study assesses the effectiveness of maternal nutrition education to reduce child stunting. (2) Methods: This was a Randomized Controlled Trial involving pairs of overweight/obese mothers with stunted children aged 2 to 5 years old in urban Indonesia. Methods: Seventy-one mother-child pairs were randomly assigned to receive either a 12-week nutrition education or printed educational materials. Mixed factorial ANOVA was used to test for between-group differences over time in relation to child’s height, weight, maternal self-efficacy, outcome expectation, and caloric intake. (3) Results: Across groups, there was a significant effect of time on child height and weight but no significant differences were observed between-groups. Maternal self-efficacy, outcome expectations in providing animal protein for the children (*p*-value = 0.025) and mother’s total caloric intake (*p*-value = 0.017) favored the intervention group over the comparison group. (4) Conclusions: The behavioral intervention produced strong improvement in maternal self-efficacy to engage in physical activity, eat fruits and vegetables and to provide children with growth-promoting animal protein, but did not significantly influence child height gain.

## 1. Introduction

In developing countries, one fourth of children under the age of five fail to grow normally because of a condition known as stunting [1]. Stunting is a condition where the child is shorter than their normal peers as measured using the height-for-age *z*-score (HAZ) of less than minus two according to the child growth standard from the WHO-Anthro 2005. Child stunting is a public health nutrition problem that hinders the development of future generations. Compared to their non-stunted peer, stunted children have shown to be more susceptible to gain more fat mass than lean mass in a cohort in Brazil [2]. After 7 to 9 years follow up, previously stunted children at 2 years of age were significantly shorter and lighter but their body mass index (BMI) or centralization of body fat was not significantly different from non-stunted South African children [3]. Beyond physiological effects, stunting may limit a child’s cognitive abilities and productivity [4]. In light of these damaging consequences, the WHO and its member countries are working to achieve a 40% reduction in child stunting by 2025 through the Scaling-Up Nutrition (SUN) program [4].

Effective community-based interventions must be developed to ameliorate child stunting and support WHO and UNICEF programs to combat child growth problems worldwide. A systematic review of the literature to explore the impact of education and complementary feeding on growth of children under 2 years of age in developing countries showed positive results [5]. In a subgroup analysis of the food secure population, child-feeding education alone yielded a significant improvement in height gain in children under the age of 2 years [5]. A previous study in Bangladesh that assessed a 3-month nutrition education intervention along with complementary feeding showed promising results for height gain [6,7]. The effect of providing complementary food and intensive nutrition education on height gain (cm) in Bangladesh was 0.80 (95% Confidence Interval (CI) = 0.007–1.53) [7]. A systematic review of community-based nutrition education programs revealed significant results when community leaders met with caregivers twice a week in their home, to deliver nutrition education programs and cooking demonstrations [8].

A demographic shift in conjunction with an epidemiological and nutrition transition has created an unusual situation in which both over- and under-nutrition occur within the same population. Child stunting is a persistent feature of this problem, known as the double burden of malnutrition. A study in a Guatemalan population informed our hypothesis that in households suffering from the double burden of malnutrition, stunted children are less likely to experience food insecurity. Results of that study revealed that the prevalence of coexistence of under-nutrition (child stunting) and over-nutrition (maternal overweight/obesity) was highest (22.7%) among those in the middle (third) quintile of socioeconomic status (SES) [9]. The study showed that maternal overweight was positively related to higher economic status while child stunting was negatively associated with higher household economic status. Lack of access to food geared to the fulfilment of dietary energy was influential for the high prevalence of child stunting but not playing the major role to double burden of malnutrition as mothers exceeded their energy consumption. Households that was suffering from double burden of malnutrition did not necessarily lacking in food access in term of energy intake. It is believed that the difference was coming from unequal food distribution in terms among household’s member. Larger number of family member or having extended family would increase the change of unmet nutrient requirement among member of the household as it varies across age groups. Top with low level of maternal nutritional literacy the problem of children having less nutrient intake resulted in their failure to grow (stunted) but the adults having excess energy intake ending up with overweight and obesity. This evidence suggests that in this socioeconomic group, relative to the others, in the absence of food insecurity and economic deprivation, modifiable factors such as food distribution and dietary diversity within the household were associated with the double burden of malnutrition. Furthermore, these households appeared to lack the capacity to direct resources properly to prevent child stunting. More specifically, we hypothesize that mothers were unable to make healthy food choices and manage food supply and distribution within the household.

A behavioral intervention was developed to target modifiable behaviors related to the double burden of malnutrition and to equip mothers with skills necessary to overcome these problems. The study was conducted in an urban setting in Indonesia through the Nutrition Education for Overweight/Obese Mother with Stunted Children (NEO-MOM) intervention. Drawing on concepts from Social Cognitive Theory (SCT), participants were prompted to set goals for themselves to improve their dietary habits and child feeding behaviors, with a focus on self-efficacy, nutrition literacy and dietary diversity.

This study was designed to test the hypothesis that for households facing the double burden of malnutrition in urban Indonesia, a behavioral intervention, coupled with a government food supplementation program, would be more effective than standard care combined with print educational materials for improving child outcomes for height and height-for-age *z*-score, maternal outcomes for weight, waist circumference, BMI, dietary diversity, dietary intake, self-efficacy, outcome expectations and nutrition literacy.

## 2. Materials and Methods

This randomized controlled trial (RCT) assessed the effectiveness of a behavioral intervention aimed at empowering mothers to address the double burden of malnutrition within the household.

### 2.1. Theory

The intervention was based on concepts of Bandura’s Social Cognitive Theory (SCT) [10], which are briefly mentioned here. Following the constructs of Social Cognitive Theory, we measured the mother’s self-efficacy, outcome expectations and knowledge measured as nutrition literacy. We developed eight measures of maternal self-efficacy, according to Bandura’s guidelines for constructing self-efficacy scales [11].

### 2.2. Sample Size and Allocation

Based on the previous study in Bangladesh [7], which found 0.8 effect size of a three months length nutrition education intervention accompanied with complementary feeding and using 90% power, a minimum of 66 total samples were required to detect changes in height gain with two tailed alphas of 0.05. At baseline this study involved 71 eligible samples that was randomly allocated to 35 in the intervention group and 36 in the comparison group/usual care. Details about the methodology and protocols of the study can be found elsewhere [12]. This study did not compare the effect of the intervention group (NEO-MOM group) with a true control group, but with a comparison group that received printed educational materials (PRINT group) plus government supplementation on child stunting and maternal overweight/obesity (see Figure 1).

### 2.3. Dietary Data

We collected two 24-h dietary food recalls per mother at set times, baseline, and at the end of intervention (three months after baseline). A portion-size guide and food models were provided to parents to assist them in estimating portion sizes. Dietary data was analyzed using NutriSurvey, a software that draws on a database containing nutrition information on typical Indonesian Food. The database is updated yearly by the Department of Nutrition, Universitas Airlangga (UA)–Indonesia. Dietary diversity was calculated following the guidance developed by the Food and Agriculture Organization of the United Nations (FAO) where mothers were asked to recall dietary consumption in the past 24 h. The answer was then aggregated into 12 food groups to create the household’s dietary diversity score (HDDS). The 12 food groups were cereals, tubers/roots, vegetables, fruits, fish, meat and poultry, eggs, nuts and seeds, dairy products, spices, oils and fats, and sweets. The dietary diversity score was ranging from 0 to 12.

### 2.4. Behavioral Measures

We measure maternal self-efficacy based on the Likert-scale ranged from 0 to 100 and covered barriers and tasks for mothers related to being physically active, to eating fruits and vegetables and providing children with animal protein in their meals. The animal protein in question was referring to any source of protein coming from animal-based products that include fish, meat and poultry, eggs, and dairy products. Outcome expectations were measured with a series of questions for the same tasks rated on a scale from 1 to 5, with 1 representing a strong disagreement and 5 representing a strong agreement. Nutrition literacy was measured in three domains: knowledge of macronutrients, skill in household food measures, and skills in grouping food in categories. There were 6-item of close-ended questions in the macronutrient’s domain, and also there were 6-item questionnaire in the household food measure domain that was reflecting the common household measurements used in Indonesia. The food groups domain was adapted from the original American “*MyPlate*” to the Indonesian version of MyPlate known in Indonesian language as “*Piring Makanku*” or recently promoted as “*Isi Piringku*”.

### 2.5. Statistical Analysis

For all variables that were normally distributed or transformed to normality, we analyzed the difference in the outcome from control and intervention group using a mixed factorial ANOVA. The within-subjects variables were the outcome variables in this research, and the between-subject variable was the group of intervention (NEO-MOM and PRINT). We used the household food insecurity access scale (HFIAS) score as covariates in the analysis. Furthermore, we conducted the ANCOVA test to see the difference in changes of primary and secondary outcome adjusted for its baseline value and the HFIAS score. For nonparametric statistics, we employed the two related-samples Mann Whitney *U* test and analyzed the data separately for the NEO-MOM group and the PRINT group with Bonferroni correction. All data analyses were performed in IBM SPSS Statistics 22 (Armonk, NY, USA). The statistical significance for all the tests was set at an alpha level of 0.05.

### 2.6. Ethical Clearance

This study was approved by The Institutional Review Board (IRB) at Kansas State University (reference number: 7894) as well as approved by the Surabaya City Review Board (Bakesbangpol No: 1366/LIT/2015) in Indonesia. All participants were explained about the study and signed the informed consent following the World Health Organization procedure, this study obtained the Universal Trial Number (UTN) U1111-1175-5834 and also registered in the Australian New Zealand Clinical Trials Registry (ANZCTR) and allocated the registration number: ACTRN12615001243505.

## 3. Results

### 3.1. Characteristics of Participants and Groups

Table 1 summarizes characteristics of the participants (children, mother, and household) at baseline in the NEO-MOM and PRINT groups. Almost all of the variables were similar and did not have significant different with the exception of child’s weight (*p*-value = 0.045) and monthly food expenditure (*p*-value = 0.010). The average of child’s weight in the NEO-MOM group was significantly lower (mean = 11.32 kg) compare to the PRINT group (mean = 12.25 kg). Monthly food expenditure in the NEO-MOM group was also around IDR 500,000 lower than those in PRINT group.

### 3.2. Intervention Effect on Outcomes and Mediators

We used the HFIAS score as covariates when testing for an intervention effect in a mixed repeated measure ANOVA.

#### 3.2.1. Child’s Outcomes

There were no significant effects observed in the group-by-linear-time trend interaction for any of the child health outcomes, but we observed a significant time effect for child weight (*p*-value = 0.023) and child height (*p*-value = 0.001). There were significant increases in weight (*p*-value < 0.001) and child height (*p*-value < 0.001) for all groups in a pairwise comparison from baseline to 3-month after baseline evaluation (Figure 2). The mean group difference for child height was 2.47 cm (95% CI = 1.55 to 3.39) and for child weight it was 0.58 kg (95% CI = 0.32 to 0.85). The ANCOVA test showed that the change in child’s height and weight was not significantly different between NEO-MOM and PRINT group (*p*-value = 0.526 and *p*-value = 0.431 respectively). In terms of child height-for-age *z*-score (HAZ), the observed improved value was not statistically significant using the related-samples Mann Whitney *U* test for both the NEO-MOM and PRINT groups (*p*-value = 0.183 and *p*-value = 0.051, respectively).

#### 3.2.2. Maternal Outcomes

There were no significant effects in the group-by-linear-time trend interaction for maternal anthropometric outcomes such as weight, waist circumference, and the BMI. Similarly, there were no significant mean group differences from baseline to three months after the baseline evaluation for maternal weight (*p*-value = 0.223), waist circumference (*p*-value = 0.929), and the BMI (*p*-value = 0.066). In this analysis, to achieve normal distribution of the data, BMI was transformed using logistic transformation. There was no significant difference in the effect of study condition for any of the maternal outcome measures. As seen in Table 2, after three months intervention, the ANCOVA test revealed that the change in mother’s weight and waist circumference was not significantly different between NEO-MOM and PRINT group (*p*-value = 0.871 and *p*-value = 0.397 respectively).

#### 3.2.3. Household Dietary Diversity

The household dietary diversity score decreased for both NEO-MOM and PRINT group after the 3-month period. In the non-parametric related samples Mann Whitney *U* test, the results showed statistical significance at Z = −2,847 (*p*-value = 0.004) and Z = −3.380 (*p*-value < 0.001). The decline in the dietary diversity score was steeper in the PRINT group (from 7.29 at baseline to 5.68 after three months) than in the NEO-MOM group (from 7.44 at baseline to 6.50 after the 3-month intervention).

#### 3.2.4. Maternal Self-Efficacy

We measured maternal self-efficacy in terms of barriers and task performance of four behaviors: being physically active, eating fruit, eating vegetables, and providing children with animal protein. All measures of maternal self-efficacy were having good internal consistency indicated by having Cronbach alpha > 0.65. The group by time interaction on maternal self-efficacy in dealing with barriers was significant for all measures, with rates of increase being more strongly positive in the NEO-MOM group than in the PRINT group (Table 3). The group by time interaction on maternal self-efficacy barriers for being physically active, eating fruits, eating vegetables, and providing the child with animal protein were all statistically significant (*p*-value = 0.030, 0.006, 0.002, and 0.042, respectively). As seen in Figure 3 the improvement in maternal self-efficacy barriers to provide their child with animal protein was in the right direction for the NEO-MOM group (from 60.36 at baseline to 67.24 after the 3-month evaluation) in contrast to the PRINT group, which showed a decrease (from 63.47 at baseline to 57.78 after a 3-month evaluation). There was a significant time effect for the maternal self-efficacy barrier of eating vegetables (*p*-value < 0.001). However, a similar time effect was not observed within subjects for the other three measures. There was no significant result in the between-subjects test. The group by linear time trend interaction effects on maternal self-efficacy to perform certain tasks was only significant for the task of eating fruit (*p*-value = 0.043) and for the task of providing animal protein for the child (*p*-value = 0.032) (See Figure 4 and Figure 5). The rate of increase in the maternal self-efficacy in the task of eating fruit was strongly positive in the intervention condition (from 49.16 at baseline to 58.19 after the 3-month evaluation) than the comparison condition, which showed a negative trend (from 50.08 at baseline to 47.66 after a 3-month evaluation).

The mother’s self-efficacy (task) in providing animal protein for their children, showed a negative trend for both the NEO-MOM and PRINT group with a steeper rate of decline in the latter group (Figure 5). In the NEO-MOM group, maternal self-efficacy (task) for providing animal protein was 63.51 at baseline and was 63.18 after the 3-month evaluation, while in the PRINT group it declined from 64.78 at baseline to 54.20 after 3-months. There was also significant time effect for maternal self-efficacy (task) for being physically active (*p*-value < 0.001). No significant between-subjects’ effects were revealed for any of the maternal self-efficacy tasks. As seen in Table 2, after three months intervention, the ANCOVA test revealed that the change in mother’s self-efficacy was shown to be significantly different between NEO-MOM and PRINT group especially in the barriers self-efficacy in eating fruit (*p*-value = 0.002), barriers self-efficacy in eating vegetables (*p*-value = 0.002), barriers self-efficacy in serving the children with animal protein (*p*-value = 0.026), task self-efficacy in eating fruit (*p*-value = 0.036), task self-efficacy in providing their children with animal protein (*p*-value = 0.039).

#### 3.2.5. Maternal Outcome Expectation

All measures of maternal outcome expectation were having good internal consistency such as: being physically active (Cronbach alpha = 0.86), eating fruit and vegetables (Cronbach alpha = 0.84), providing animal protein for their kids (Cronbach alpha = 0.89). Measures of maternal outcome expectation were not normally distributed. Even though all measures showed positive increase in the NEO-MOM group relative to the PRINT group (Table 4), there was only one measure, providing animal protein for the child that increased significantly in the NEO-MOM group (Z = −2.242; *p*-value = 0.025). Maternal outcome expectation for providing the children with animal protein was increasing for both NEO-MOM group (from 5.12 at baseline to 5.33 at 3-month evaluation) and PRINT group (from 5.13 at baseline to 5.17 at three months) even though it was not statistically significant.

#### 3.2.6. Maternal Nutrition Literacy

All measures of maternal nutrition literacy were not showing good internal consistency with Cronbach alpha <0.65. At baseline, Cronbach alpha obtained for maternal nutrition literacy for macronutrient domain was 0.56, for household food measures was 0.14, and for grouping foods according to Indonesian version of MyPlate was 0.41. After three months intervention, the internal consistency was also not good for all domains of nutrition literacy: macronutrient (Cronbach alpha = 0.53), household food measures (Cronbach alpha = 0.33), grouping foods (Cronbach alpha = 0.64).

Because the data for maternal literacy was not normally distributed, we employed repeated Mann Whitney test for statistical analysis. Results showed no significant effect of the intervention on the mother’s nutrition literacy measures (Table 4). The greatest change in the nutrition literacy was observed in the NEO-MOM group for the literacy test for food group categorization test using the Indonesian version of MyPlate called “*Piring Makanku*” (Z = −1.442; *p*-value = 0.149).

#### 3.2.7. Maternal Dietary Intake

Based on the results of the Mann Whitney test, almost all measures of maternal dietary intake showed no significant effect with only total energy (caloric) intake that was statistically significant in the NEO-MOM group (Table 4). Mother’s total energy intake in the NEO-MOM group decreased from 1075 kcal at baseline to 845 kcal at 3 months (Z = −2.393; *p*-value = 0.017) and declined in the PRINT group from 1029 kcal at baseline to 840 at 3 months (Z = −1.135; *p*-value = 0.257).

#### 3.2.8. Moderation of Intervention Effects

For all variables that passed the normality assumption we used the household food insecurity access scale (HFIAS) score as the covariate. Our results showed that the HFIAS score was a significant moderator of some of the significant outcome variables measures. Treated as a covariate in the mixed method ANOVA analysis, the HFIAS score revealed significant between-subjects effects for maternal self-efficacy (barriers) in providing children with animal protein (F(1, 64) = 5.534, *p*-value = 0.022), self-efficacy (task) in eating fruit (F(1, 64) = 4.943, *p*-value = 0.030), self-efficacy (task) in eating vegetables (F(1, 64) = 4.781, *p*-value = 0.033), and self-efficacy (task) in providing their child with animal protein (F(1, 64) = 6.802, *p*-value = 0.011).

## 4. Discussion

The goal of this study was to empower and equip overweight or obese mothers to overcome the double burden of malnutrition. Mothers received training on strategies in overcoming the double burden of malnutrition. They were trained through behavioral strategies such as mastery experience, vicarious experience, goal setting, and verbal motivation to achieve better health outcomes for themselves, as well as for their children. The hypothesis of this randomized controlled trial was that applying behavioral intervention strategies based on Bandura’s Social Cognitive Theory for three months would be effective in improving child growth to address the issue of child stunting in a household facing the double burden of malnutrition. Results revealed that after a 3-month intervention there was a positive increase in child height in both groups but no catch-up in terms of the HAZ score. The lack of significance in the time and group interactions as well as the test of the between-subjects effect indicated that the significant time effect observed could well be attributable to the natural growth of the child and not the intervention.

Another possible explanation for the observed results might be related to the compliance of the mothers in implementing the Indonesian government food supplementation program for the underweight children. Compared to the previous study in Bangladesh that included food supplementation as a part of the intervention [6,7], in our study we relied on the food supplementation part from the government program and use it as an inclusion criterion for eligible participants. Hence, the amount of effort in ensuring participant’s compliance in the use of food supplementation in the Bangladeshi study was likely to be more rigorous than this one. But, because we did not measure the compliance rate related to the consumption of food supplementation from the government, a comparison with previous study in Bangladesh was not possible.

Furthermore, children 2 to 5 years old have passed the optimum time for rapid linear growth that occurs during the first 1000 days of life. However, stunted children in this age range needed an intervention to catch up with their normal peers. In terms of the height-for-age *z*-score (HAZ) measure, we did not find a significant effect in both the intervention and comparison group. The fact that our intervention targeted children from the age of 24 months to less than 5 years old might have hindered the effect in comparison to one targeting children during the first 1000 days of life or from the womb up to 24 months old when they have the best opportunity for growth improvement. Results of a meta-analysis on interventions aimed at improving child nutritional status revealed that interventions were generally more effective for children under the age of 2 years, and for those who were nutritionally deprived [13].

We saw a significant increase in child’s weight overtime for both group (*p*-value = 0.023), and there was borderline significant difference in the weight change from baseline between intervention and comparison group (*p*-value = 0.050). While for child’s height, no significant difference was observed between the effect of intervention and comparison condition (*p*-value = 0.143). This supports a previous study in a developing country that reported greater effect size in increasing a child’s weight than in improving a child’s linear growth [6].

All of the maternal primary outcome measures showed no significant improvement after a 3-month intervention for both groups. For anthropometric outcomes such as weight, waist circumference and BMI it might take longer for the intervention to have significant effects. The length of time employed in the current study was calculated based on the time needed to improve child’s height from a previous study and was not based on the maternal anthropometric measures. Therefore, the lack of significant effects in our study for maternal primary outcomes might be due to the insufficient length of the intervention.

However, we saw significant improvement on almost all measures of maternal self-efficacy for both tasks and barriers. There was significant increase for all four maternal self-efficacy (barriers) and two of the maternal self-efficacy (tasks) overtime. This result aligned with a study in Australia that showed maternal self-efficacy was a good predictor for the quality of diet among children aged 3 to 5 years [14]. Mother’s child feeding behavior was indirectly related to child vegetable intake through maternal feeding self-efficacy in an Australian population [15]. Results from qualitative studies also support the importance of self-efficacy in influencing healthy eating decision-making [16] and food preparation behavior [17]. Even though we did not directly measure mother’s behavior towards child feeding practices, we found significant effect of the intervention for both NEO-MOM and PRINT group in household’s dietary diversity as an indirect measure of maternal healthy food choice behavior.

We measured behavior in terms of dietary diversity as an indication of healthy food choice in the household. The results revealed a significant effect but in a negative direction. After the 3-month intervention, the average household dietary diversity score significantly decreased for both the NEO-MOM group and comparison group. This may have been affected by the time the interview was conducted between baseline and the 3-month evaluation. Baseline data were collected at the beginning of the month, while the evaluation data was collected at the end of the month. Results may have been influenced by food budget availability and the fact that most Indonesian people receive their salary at the beginning of the month and may have had more money to spend on food at that time, as compared to the end of the month [18].

The effect of our intervention on maternal dietary intake was significant only for total energy (caloric) intake in the intervention group. These results align with previous RCTs on weight loss that suggest that as the first step in trying to lose weight, women tend to reduce their caloric intake. Other nutrient intake did not show significant results, perhaps because our two times in 24 h dietary recall may not have been sufficient to capture the variability of micronutrient intake as compared to total energy [19].

There was no significant effect revealed in the three domains of maternal nutrition literacy used in this study. These results might be related to the fact that our intervention was designed based on the behavioral strategies that followed the tenets of Social Cognitive Theory. Even though we provided six weeks (over 600 h) of nutrition education classes to improve mothers’ knowledge, the content might not necessarily fit with the questions included in the validated nutrition literacy questionnaire [20]. The tools for nutrition literacy developed in the more highly educated settings in the U.S. might be too difficult to for mothers with less education in developing countries such as Indonesia.

### Strengths and Limitation

To the best of our knowledge, this study was the first to conduct a randomized controlled trial (RCT) on households experiencing the problem of double burden of malnutrition in the form of the coexistence of stunted children and overweight/obese mother pairs (SCOWT). The strength of the study was a solid methodological approach and RCT design, a small attrition rate of around 5.5% in each group, and the high participation from local community health workers to deliver the intervention that promote adoption of our strategies. However, with limited resources, we could not incorporate supplementary feeding as part of our intervention, but we made use of the Indonesian government’s 3-month supplementary feeding program to overcome severely underweight children as inclusion criteria. Therefore, a limitation of the study was the absence of a true control group. Without it, it is impossible to know whether the observed time effect was significantly different from natural growth following the age increase. For this reason, we may have underestimated the effect of print educational materials and ongoing food supplementation by the Indonesian government. Our observed effect could have been higher if we did not use the HFIAS score (measure of food insecurity) as covariates in the analysis. In this study we also assume that the mother was the focal member of the household responsible for purchasing food and its distribution in the family, which might not always be true. Other concerns might arise from the use of relatively high effect size from previous study [7] in calculating the sample size that might be attributable to under power the study. However, we minimize the effect by using fairly substantial bigger power relatively compared to the traditional 80%. The application of our results might be limited to an urban population in a developing country setting.

## 5. Conclusions

The behavioral-based nutrition education intervention produced strong improvement in maternal self-efficacy to engage in physical activity, eat fruits and vegetables, and to provide children with growth-promoting animal protein, but did not significantly influence child height gain. Although both of our interventions (NEO-MOM and PRINT) allowed significant increases in child growth overtime, no catch-up growth was observed in either group. Relative to the PRINT comparison group, our intervention improved almost all maternal self-efficacy measures, which are viewed as necessary steps for engaging in healthy behaviors. The behavioral intervention in this study was deemed feasible and it had a good retention rate. This study provides a basis for potential strategies to reduce the rate of child stunting in households undergoing double burden of malnutrition.

## Figures and Tables

**Figure 1 nutrients-10-01910-f001:**
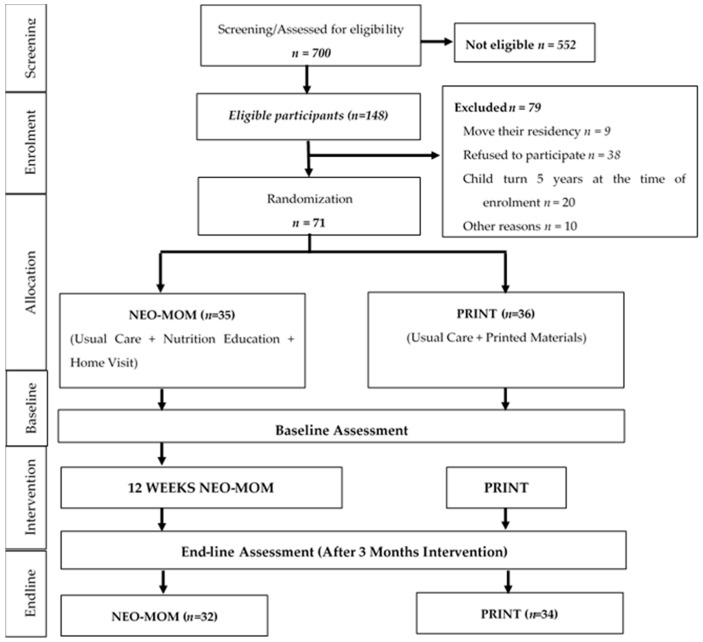
Adapted CONSORT diagram of the Nutrition Education for Overweight/Obese Mother with Stunted Children (NEO-MOM) study.

**Figure 2 nutrients-10-01910-f002:**
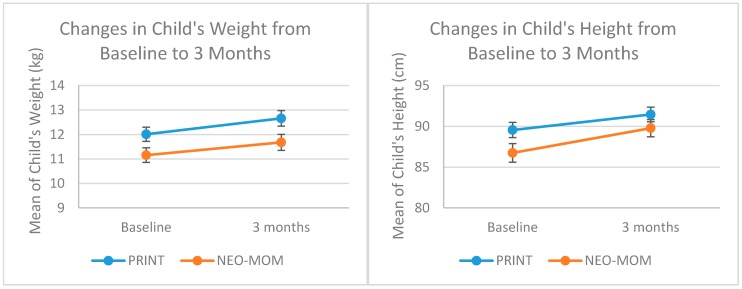
Profile plot of child’s weight and height change from baseline to three months evaluation.

**Figure 3 nutrients-10-01910-f003:**
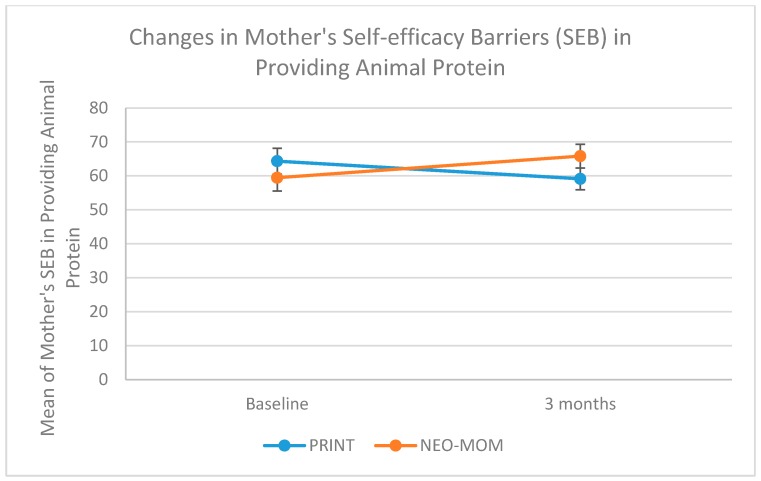
Profile plot of maternal self-efficacy (barriers) for providing animal protein.

**Figure 4 nutrients-10-01910-f004:**
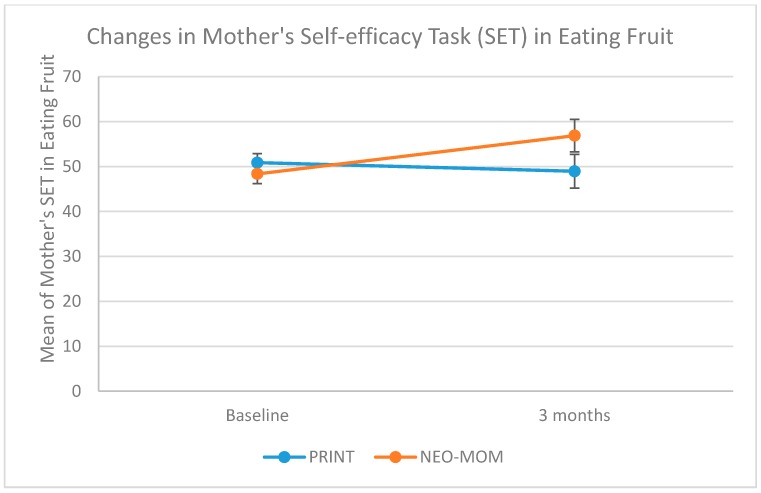
Profile plot of maternal self-efficacy (task) for eating fruit.

**Figure 5 nutrients-10-01910-f005:**
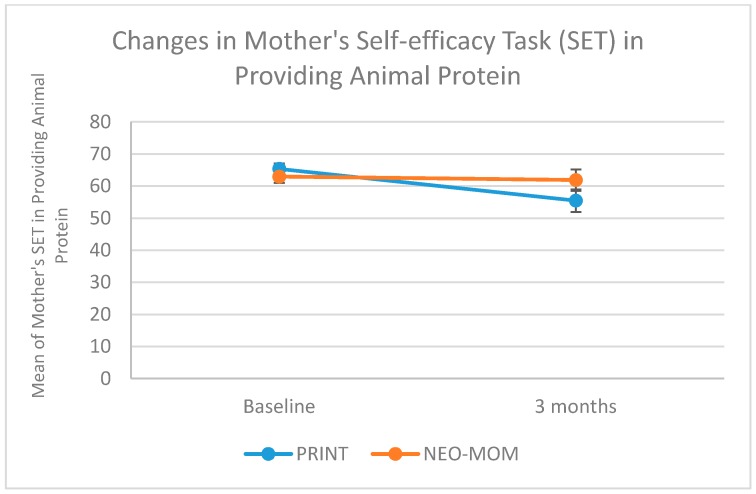
Profile plot of maternal self-efficacy (task) for providing animal protein.

**Table 1 nutrients-10-01910-t001:** Participants characteristics at baseline (*n* = 71).

Variable	NEO-MOM (*n* = 35)		PRINT (*n* = 36)		*p*-Value
	Mean	(SD)	Mean	(SD)	
**Child characteristics**					
Age (months)	39.57	7.82	40.24	8.11	0.679
Weight (kg)	11.32	1.92	12.25	2.44	0.045 *
Height (cm)	86.84	6.34	89.43	5.34	0.060
Height-for-age *Z*-score	−2.998	0.85	−2.674	0.57	0.071
**Maternal characteristics**					
Age (years)	34.09	6.86	31.47	6.76	0.123
Education (years)	7.29	7.75	7.89	3.62	0.417
Weight (kg)	64.59	8.83	67.99	10.15	0.233
Height (cm)	147.43	5.11	148.23	4.67	0.441
Waist circumference (cm)	92.61	8.30	93.91	9.72	0.364
**Household characteristics**					
Dietary diversity score	7.29	1.86	7.22	1.80	0.737
HFIAS score	8.94	5.75	5.92	5.49	0.061
Monthly Income (IDR)	1,532,857	512,769	2,116,666	1,448,562	0.054
Monthly Food expenditure (IDR)	974,074	343,726	1,217,307	316,418	0.010 *
**Maternal physical activity**					
Average daily step	3156	2134	2899	2356	0.196

Note. Significance based on α = 0.05; * *p*-value < 0.05. The analysis is based on Independent *t*-test.

**Table 2 nutrients-10-01910-t002:** The ANCOVA test results on primary outcomes & maternal self-efficacy (*n* = 66).

Variable	F	*p*-Value	Partial Eta Squared	Adjusted R Squared
**Child’s outcomes**				
Weight (kg)	0.629	0.431	0.010	0.151
Height (cm)	0.407	0.526	0.007	−0.022
**Maternal outcomes**				
Weight (kg)	0.027	0.871	0.000	−0.028
Waist circumference (mm)	0.726	0.397	0.012	−0.025
BMI ^#^	0.115	0.736	0.002	−0.023
**Maternal self-efficacy (Barrier)**				
Being physically active	2.035	0.159	0.032	0.323
Eating fruit	10.011	0.002 *	0.139	0.404
Eating vegetables	10.238	0.002 *	0.142	0.236
Providing animal protein for kids	5.224	0.026 *	0.078	0.474
**Maternal self-efficacy (Task)**				
Being physically active	3.922	0.052	0.059	0.276
Eating fruit	4.624	0.036 *	0.070	0.096
Eating vegetables	3.137	0.081	0.048	0.179
Providing animal protein for kids	4.468	0.039 *	0.067	0.081

*Note*. Significance based on α = 0.05; * *p*-value < 0.05. ^#^ The analysis is based on log-transformed variable.

**Table 3 nutrients-10-01910-t003:** Group means and test of within- and between- subject effect (*n* = 66).

Outcome/Mediator	Mean (SE)				Test of Within-Subject Effect		Test of Between-Subject Effect
	Intervention (*n* = 32)		Comparison (*n* = 34)		Partial Eta Square (*p*)		
	Baseline	3 Month	Baseline	3 Month	Time	Time by Group Interaction	Group Difference
**Child’s outcomes**							
Weight (kg)	11.19 (0.31)	11.69 (0.34)	11.98 (0.29)	12.64 (0.33)	0.080 (0.023) *	0.005 (0.578)	0.059 (0.050) *
Height (cm)	86.80 (1.07)	89.90 (1.01)	89.49 (1.04)	91.34 (0.98)	0.173 (0.001) **	0.027 (0.189)	0.034 (0.143)
**Maternal outcomes**							
Weight (kg)	65.17 (1.49)	64.87 (1.54)	68.35 (1.44)	67.91 (1.49)	0.034 (0.139)	0.001 (0.819)	0.033 (0.146)
Waist circumference (mm)	92.18 (1.60)	91.73 (1.80)	95.08 (1.56)	95.67 (1.74)	0.000 (0.992)	0.008 (0.505)	0.034 (0.141)
BMI ^a^	30.13 (0.62)	29.95 (0.59)	31.01 (0.66)	30.63 (0.75)	0.054 (0.062)	0.006 (0.543)	0.022 (0.235)
**Maternal self-efficacy (Barrier)**							
Being physically active			50.55 (2.53)				
Eating fruit	43.95 (2.61)	49.11 (2.88)	55.66 (3.69)	45.34 (2.79)	0.008 (0.472)	0.072 (0.030) *	0.003 (0.650)
Eating vegetables	53.04 (3.80)	62.49 (3.41)	38.03 (2.54)	49.82 (3.30)	0.025 (0.207)	0.115 (0.006) *	0.021 (0.251)
Providing animal	35.29 (2.62)	61.13 (3.39)	63.47 (3.83)	48.54 (3.28)	0.376 (<0.001) ***	0.146 (0.002) **	0.029 (0.174)
protein for kids	60.36 (3.95)	67.24 (3.31)		57.78 (3.21)	0.008 (0.491)	0.064 (0.042) *	0.009 (0.448)
**Maternal self-efficacy (Task)**							
Being physically active	34.71 (3.99)	56.19 (4.17)	30.53 (3.86)	43.62 (4.04)			
Eating fruit	49.16 (2.12)	58.19 (3.73)	50.08 (2.09)	47.66 (3.67)	0.203 (<0.001) ***	0.026 (0.199)	0.047 (0.083)
Eating vegetables	55.17 (2.19)	58.57 (3.48)	55.30 (2.12)	49.98 (3.38)	0.028 (0.185)	0.071 (0.034) *	0.031 (0.167)
Providing animal	63.51 (1.83)	63.18 (3.42)	64.78 (1.78)	54.20 (3.32)	0.014 (0.355)	0.041 (0.105)	0.028 (0.182)
protein for kids					0.002 (0.717)	0.071 (0.032) *	0.024 (0.217)

Note. Significance based on α = 0.05; *** *p*-value < 0.001, ** *p*-value < 0.01, * *p*-value < 0.05. ^a^ The analysis is based on log-transformed variable.

**Table 4 nutrients-10-01910-t004:** Group means and differences between group means for all outcomes in non-parametric statistics (*n* = 66).

Outcome/Mediator	Intervention				Comparison			
	Mean (SD)		Z	*p*-Value	Mean (SD)		Z	*p*-Value
	Baseline (*n* = 32)	3 Month (*n* = 32)			Baseline (*n* = 34)	3 Month (*n* = 34)		
**Child’s HAZ**	−2.99 (0.85)	−2.85 (0.79)	−1.333	0.183	−2.67 (0.57)	−2.55 (0.61)	−1.951	0.051
**Mother’s BMI**	30.13 (3.52)	29.95 (3.34)	−0.895	0.371	31.01 (3.84)	30.63 (4.36)	−1.646	0.100
**Household dietary diversity**	7.44 (1.70)	6.50 (2.11)	−2.847	0.004 *	7.29 (1.75)	5.68 (1.61)	−3.380	<0.001 **
**Maternal outcome expectation**								
Being physically active	5.04 (0.65)	5.21 (0.48)	−1.381	0.167	5.01 (0.72)	4.86 (0.72)	−1.082	0.279
Eating fruit & vegetables	5.06 (0.56)	5.18 (0.49)	−1.312	0.190	4.98 (0.54)	4.97 (0.57)	−0.152	0.879
Providing animal protein for kids	5.12 (0.54)	5.33 (0.48)	−2.242	0.025 *	5.13 (0.59)	5.17 (0.56)	−0.320	0.749
**Maternal nutrition literacy**								
Macronutrient	2.63 (1.62)	2.47 (1.48)	-0.630	0.529	3.15 (1.50)	3.03 (1.62)	−0.367	0.714
Household food measures	1.78 (1.01)	1.75 (0.76)	−0.339	0.735	1.53 (1.11)	1.32 (0.98)	−0.920	0.358
MyPlate categorization	12.72 (1.78)	13.19 (2.39)	−1.442	0.149	12.79 (1.90)	12.82 (1.96)	−0.039	0.969
**Maternal dietary intake**								
Energy	1075 (538)	845 (559)	−2.393	0.017 *	1029 (774)	840 (502)	−1.135	0.257
Protein	57.29 (42.11)	44.69 (39.59)	−1.627	0.104	46.50 (38.99)	39.28 (32.47)	−1.027	0.304
Fat	49.95 (51.99)	37.82 (48.80)	−1.646	0.100	56.21 (62.00)	43.29 (48.23)	−0.652	0.514
Carbohydrate	103.18 (74.50)	90.75 (69.79)	−0.926	0.355	91.71 (59.06)	77.71 (39.49)	−1.349	0.177
Iron	6.62 (8.04)	6.58 (11.54)	−1.197	0.231	6.32 (7.69)	5.22 (3.40)	−0.527	0.598
Zinc	3.93 (2.13)	3.18 (2.27)	−1.833	0.067	3.62 (2.51)	3.35 (1.89)	−0.225	0.822
Calcium	140.17 (96.21)	144.31 (200.88)	−1.141	0.254	157.05 (190.80)	159.81 (186.58)	−0.527	0.698
Vitamin A	842 (559)	680 (1466)	−1.496	0.135	982 (2441)	503 (738)	−1.930	0.054
Fiber	6.77 (7.29)	6.75 (10.60)	−1.029	0.303	8.47 (14.88)	5.45 (5.22)	−1.524	0.127

Note. The test was based on repeated measures non-parametric statistic of Mann Whitney U. Note. Significance based on α = 0.05; ** *p*-value < 0.001, * *p*-value < 0.05.

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
