# Peer review of "The Effectiveness of Nutrition Education for Overweight/Obese Mother with Stunted Children (NEO-MOM) in Reducing the Double Burden of Malnutrition"

_nutrients, 2018, doi:10.3390/nu10121910_

Round 1
Reviewer 1 Report
This paper is straightforward and generally well written.
My only comments relate to making the findings in terms of stunting more explicit.
In the discussion the results would be better expressed by the following addition after the sentence (line 292) Results revealed that after a 3-month intervention there was a positive increase in child height in both groups but no catch-up in terms of the HAZscore.
Also in the Conclusions (382) the results would be better expressed by the additions in the sentence In conclusion, although both of our interventions (NEO-MOM and PRINT) allowed significant increases in child growth overtime, no catch-up growth was observed in either group.
Author Response
This paper is straightforward and generally well written.
Thank you for your review
My only comments relate to making the findings in terms of stunting more explicit.
In the discussion the results would be better expressed by the following addition after the sentence (line 292) Results revealed that after a 3-month intervention there was a positive increase in child height in both groups but no catch-up in terms of the HAZscore.
Thank you for your input, we add the sentence in the discussion as suggested.
Also, in the Conclusions (382) the results would be better expressed by the additions in the sentence in conclusion, although both of our interventions (NEO-MOM and PRINT) allowed significant increases in child growth overtime, no catch-up growth was observed in either group.
Thank you for your input, we add the suggested sentence in the conclusion.
Reviewer 2 Report
The theoretical basis for the study is well explained. The methods section should be restructured to make it easier to follow and requires more detail/clarification on the outcome measures. The results section is too lengthy and could be made more concise.The discussion is well written. There are some minor English language errors.
Suggested changes:
Abstract, line 24: Would read better as: "This study assess the effectiveness of maternal nutrition eduction to reduce child stunting"
Abstract, line 30: Change to: across "groups"
Introduction, line 42: Please give a definition of what stunting is and how it is measured. Readers may not be familiar with the precise definition.
Line 44: Change to "Compared to their non-stunted peer"
Line 75: "we suggest that mothers were unable to make healthy food choices" : what is the role of paternal weight status/diet on stunted children? What is the role of the extended family/community in this population? What are the cultural/socioeconomic issues around food scarcity/availability? These factors could be mentioned in the introduction and added as a limitation/clarification in the discussion section.
Materials and Methods section lines 90-134
This section in difficult to follow. Suggest restructure it with several subheadings (e.g. theory, participants, inclusion/exclusion criteria, recruitment, sample size, dietary data, self efficacy, statistical methods, ethical considerations etc).
Please clarify how dietary diversity is calculated and what is meant by "animal protein"? Does this only include meat/fish ? eggs? dairy products? How is nutrition literacy defined/measured?
Results line 181-246 Maternal self efficacy & outcome expectations: Suggest this section should be reduced, it is too long and the key results are lost as there is too much text. For example, it is not necessary to report all the Cronbach Alpha scores.
Conclusions lines 381-388: The conclusion in lines 381-388 does not particularly align with the conclusion in the abstract, which states that "the behavioural intervention.......did not significantly influence child health gain". This key message should be incorporated into the main conclusion of the article also.
Line 385:"the study was well received by participants": was this formally assessed? If not, it should not be included. Instead you could comment that the intervention was feasible and/or had a good retention rate instead.
Author Response
Reviewer 2 Comments
Open Review
(x) I would not like to sign my review report
( ) I would like to sign my review report
English language and style
( ) Extensive editing of English language and style required
(x) Moderate English changes required
( ) English language and style are fine/minor spell check required
( ) I don't feel qualified to judge about the English language and style
Yes | Can be improved | Must be improved | Not applicable | |
Does the introduction provide sufficient background and include all relevant references? | ( ) | (x) | ( ) | ( ) |
Is the research design appropriate? | (x) | ( ) | ( ) | ( ) |
Are the methods adequately described? | ( ) | (x) | ( ) | ( ) |
Are the results clearly presented? | (x) | ( ) | ( ) | ( ) |
Are the conclusions supported by the results? | ( ) | (x) | ( ) | ( ) |
Comments and Suggestions for Authors
The theoretical basis for the study is well explained. The methods section should be restructured to make it easier to follow and requires more detail/clarification on the outcome measures. The results section is too lengthy and could be made more concise. The discussion is well written. There are some minor English language errors.
Thank you for your thorough review and suggestions. We are revising it accordingly.
Suggested changes:
Abstract, line 24: Would read better as: "This study assess the effectiveness of maternal nutrition education to reduce child stunting"
Thank you for your input, we revise it as suggested.
Abstract, line 30: Change to: across "groups"
Thank you for your input, we change it to: across “groups”.
Introduction, line 42: Please give a definition of what stunting is and how it is measured. Readers may not be familiar with the precise definition.
Thank you for your suggestion, we add the definition as follows: “Stunting is a condition where the child is shorter than their normal peers as measured using the height-for-age z-score (HAZ) of less than minus two according to the child growth standard from the WHO-Anthro 2005.”
Line 44: Change to "Compared to their non-stunted peer"
Thank you for your input, we change it as suggested.
Line 75: "we suggest that mothers were unable to make healthy food choices" : what is the role of paternal weight status/diet on stunted children? What is the role of the extended family/community in this population? What are the cultural/socioeconomic issues around food scarcity/availability? These factors could be mentioned in the introduction and added as a limitation/clarification in the discussion section.
Thank you for your input, we add several sentences in the introduction section as well as in the limitation sub-section.
In the introduction we add:
“The study showed that maternal overweight was positively related to higher economic status while child stunting was negatively associated with higher household economic status. The finding from the Guatemalan study indicated that lack of access to food geared to the fulfilment of dietary energy was influential for the high prevalence of child stunting but not playing the major role to double burden of malnutrition as mothers exceeded their energy consumption. Household that was suffering from double burden of malnutrition did not necessarily lacking in food access in term of energy intake. It is believed that the difference was coming from unequal food distribution in terms among household’s member. Larger number of family member or having extended family would increase the change of unmet nutrient requirement among member of the household as it varies across age groups. Top with low level of maternal nutritional literacy the problem of children having less nutrient intake resulted in their failure to grow (stunted) but the adults having excess energy intake ending up with overweight and obesity.”
In the limitation we add:
“In this study we also assume that the mother was the focal member of the household responsible for purchasing food and its distribution in the family which might not always be true.”
Materials and Methods section lines 90-134
This section in difficult to follow. Suggest restructure it with several subheadings (e.g. theory, participants, inclusion/exclusion criteria, recruitment, sample size, dietary data, self-efficacy, statistical methods, ethical considerations etc).
Thank you for your suggestion, we restructure the material and methods section into several subheadings.
Please clarify how dietary diversity is calculated and what is meant by "animal protein"? Does this only include meat/fish? eggs? dairy products? How is nutrition literacy defined/measured?
Thank you for your suggestion, we add more explanation related to dietary diversity, “animal protein” and nutrition literacy in the materials and methods section lines restructure the material and methods section into several subheadings.
“Dietary diversity was calculated following the guidance developed by the the Food and Agriculture Organization of the United Nations where mothers were asked to recall dietary consumption in the past 24 hours. The answer was then aggregated into 12 food groups to create the household’s dietary diversity score (HDDS). The 12 food groups were cereals, tubers/roots, vegetables, fruits, fish, meat and poultry, eggs, nuts and seeds, dairy products, spices, oils and fats, and sweets. The dietary diversity score was ranging from 0 to 12.”
“Animal protein in question was referring to any source of protein coming from animal-based products that include fish, meat and poultry, eggs, and dairy products.”
“Nutrition literacy was measured in three domains: knowledge of macronutrients, skill in household food measures, and skills in grouping food in categories. There were 6-item of close-ended questions in the macronutrient’s domain, and also there were 6-item questionnaire in the household food measure domain that was reflecting the common household measurements used in Indonesia. The food groups domain was adapted from the original American “MyPlate” to the Indonesian version of MyPlate known in Indonesian language as “Piring Makanku” or “Isi Piringku”.
Results line 181-246 Maternal self-efficacy & outcome expectations: Suggest this section should be reduced, it is too long and the key results are lost as there is too much text. For example, it is not necessary to report all the Cronbach Alpha scores.
Thank you for your suggestion, we reduce it accordingly.
Conclusions lines 381-388: The conclusion in lines 381-388 does not particularly align with the conclusion in the abstract, which states that "the behavioural intervention.......did not significantly influence child health gain". This key message should be incorporated into the main conclusion of the article also.
Thank you for your suggestion, we incorporated the key message as stated in the conclusion of the abstract into the main conclusion of the article.
Line 385:"the study was well received by participants": was this formally assessed? If not, it should not be included. Instead you could comment that the intervention was feasible and/or had a good retention rate instead.
No, it was not formally assessed. We delete it and add the comment that the behavioural intervention in this study was deemed feasible and it had a good retention rate.